# Gait smoothness during high-demand motor walking tasks in older adults with mild cognitive impairment

Thanpidcha Poosri[1], Sirinun Boripuntakul[1]*, Somporn Sungkarat[1], Teerawat Kamnardsiri[2], Atiwat Soontornpun[3], Kanokporn Pinyopornpanish[4]

1 Faculty of Associated Medical Sciences, Department of Physical Therapy, Chiang Mai University, Chiang Mai, Thailand, 2 Department of Digital Game, College of Arts, Media and Technology, Chiang Mai University, Chiang Mai, Thailand, 3 Faculty of Medicine, Department of Internal Medicine, Chiang Mai University and Northern Neuroscience Center, Chiang Mai, Thailand, 4 Faculty of Medicine, Department of Family Medicine, Chiang Mai University, Chiang Mai, Thailand

* sirinun.b@cmu.ac.th

**Data Availability Statement:** All relevant data are within the manuscript and its Supporting Information files.

## Abstract

Early signs of Mild Cognitive Impairment (MCI)-related gait deficits may be detected through the performance of complex walking tasks that require high gait control. Gait smoothness is a robust metric of overall body stability during walking. This study aimed to explore gait smoothness during complex walking tasks in older adults with and without MCI. Participants were 18 older adults with MCI (mean age = 67.89 ± 4.64 years) and 18 cognitively intact controls (mean age = 67.72 ± 4.63 years). Gait assessment was conducted under four complex walking tasks: walking a narrow path, walking around an obstacle, horizontal head turns while walking, and vertical head turns while walking. The index of harmonicity (IH), representing gait smoothness associated with overall body stability, was measured in anteroposterior, mediolateral, and vertical directions. A multivariate analysis was employed to compare the differences in IH between groups for each complex walking task. The MCI group demonstrated a reduction of IH in the mediolateral direction during the horizontal head turns than the control group (MCI group = 0.64 ± 0.16, Control group = 0.74 ± 0.12, p = 0.04). No significant differences between groups were found for the IH in other directions or walking conditions. These preliminary findings indicate that older adults with MCI have a decline in step regularity in the mediolateral direction during walking with horizontal head turns. Assessment of the smoothness of walking during head turns may be a useful approach to identifying subtle gait alterations in older adults with MCI, which may facilitate timely gait intervention.

## Introduction

Higher level cognitive function, specifically executive function, and attention, plays a vital role in integrating sensorimotor information and controlling movement execution, contributing to safe mobility in complex real-life environments [1,2]. Previous studies have found that

**Funding:** -Initials of the authors who received each award: Thanpidcha Poosri -Grant numbers awarded to each author: AMS-2564 -The full name of each funder: Faculty of Associated Medical Sciences, Chiang Mai University, Thailand. -URL of each funder website: N/A -Did the sponsors or funders play any role in the study design, data collection and analysis, decision to publish, or preparation of the manuscript?: The funders had no role in study design, data collection and analysis, decision to publish, or preparation of the manuscript.

**Competing interests:** The authors have declared that no competing interests exist.

individuals with Mild Cognitive Impairment (MCI), a clinical entity involving cognitive deficit beyond normal aging [3,4], displayed a decline in attention and executive function [5] which may in turn result in gait abnormalities and increased risk of falling [6,7]. Therefore, the early identification of subtle MCI-related changes in gait may have important clinical benefits, ensuring a timely assessment and intervention in gait-related concerns.

Existing studies have demonstrated that simple steady-state walking tasks allow the detection of gait changes related to falls in older adults with MCI compared to their cognitively intact peers [4,8,9]. While the above findings were promising, some studies failed to find differences in gait features between persons with MCI and those cognitively intact controls [4,9,10]. Discrepancies across prior findings may be due to the variations in gait tasks and outcome measures. It is posited that complex walking tasks could exacerbate subtle MCI-related gait impairment. Currently, it is well documented that gait impairment of individuals with MCI is significantly pronounced when adding a cognitive load while walking (dual-tasking) [9]. However, previous studies examining MCI-related changes in gait under motor load tasks relevant to a real-life walking situation are scarce. Limited studies have demonstrated a decline in gait regulation of older adults with MCI during transitional gait speeds, including gait initiation, termination, and slow speed transition [11–13]. These findings suggest that motor load tasks may be sensitive in detecting subtle gait changes in older adults with MCI.

Head turning is a fundamental component for successful responses to environmental demands via visual scanning, such as checking for traffic before crossing the road and navigating the barriers in the travel path [14]. Basically, head turning while walking alters visual and vestibular information from the changes of visual field. Specifically, when individuals need to adjust their gait in real-time while head-turning in response to environmental constraints, the vestibular system is more burdened with tracing head movements, which may compromise gait stability [15]. Previous studies have elicited a reduction in postural stability in individuals with MCI partly due to a decrement in the vestibular system linked to a decline in hippocampus function [16]. Therefore, it is plausible that head manipulation while walking could be considered a candidate measure of gait stability in older adults with MCI. Moreover, complex walking tasks concerning everyday environments that require high motor demands in older adults, including walking a narrow path and around an obstacle [17,18], may challenge gait control in older adults with MCI. Taken together, high-demand motor walking tasks, including head turns while walking, walking a narrow path, and walking around an obstacle, may be used as a motor stress test to amplify a subtle change of gait in older adults with MCI.

The smoothness of walking, the rhythmic pattern of trunk acceleration, is a dynamic gait metric useful to quantify gait regularity during walking [19–21]. Prior works have demonstrated impaired gait smoothness across various populations, such as cognitively impaired individuals, older adults with restricted neuromuscular control, Parkinson's disease patients, and individuals with sensory impairment, compared to young adults and persons without the disease [21–27]. According to recent studies, trunk accelerations are more sensitive than traditional spatiotemporal metrics for capturing subtle alterations in gait [26,28]. Among gait smoothness measures, the index of harmonicity (IH) provides information on the harmonicity of trunk and pelvis movement in conjunction with gait rhythm through spectral analysis of trunk acceleration signals [20]. Specifically, the IH indicates the smoothness of rhythmic intersegmental coordination between the trunk and pelvis throughout the gait cycle, providing insight into balance control during walking [21]. It is well established that dynamic balance and movement coordination during walking are prerequisites for gait stability. With this, the IH has been proposed as a marker indicating the smoothness of gait in relation to walking stability.

To date, gait smoothness has been used to examine gait quality in limited groups and walking conditions. In addition, to the best of our knowledge, no study has investigated the smoothness of gait in people with MCI under high-demand motor tasks relevant to real-life environmental conditions. The aim of this preliminary study was to explore subtle MCI-related changes in gait under a high-demand motor walking task by using gait smoothness as a measurement indicator. It was hypothesized that the MCI group would have less gait smoothness under motor load tasks than the control group. Findings in this study might offer early detection of gait deficits and facilitate timely gait intervention for older adults with MCI.

## Material and methods

### Participants

The sample size was calculated based on our pilot study, in which the smallest effect size was chosen. The smallest effect size of 0.22 was derived from the IH in the anteroposterior direction in the condition of walking while making vertical head turns (with baseline gait speed as a covariate). With this, a general linear model with an effect size of 0.22, a power of 0.80, and an alpha level of 0.05 revealed a total sample size of at least 26 participants (13 participants per group).

Eighteen community-dwelling older adults with MCI and 18 healthy-matched controls aged 60 or older were enrolled in the present study. The older adults with MCI were recruited from Maharaj Nakorn Chiang Mai Hospital, a tertiary care hospital in Northern Thailand. The diagnosis of MCI was performed by an experienced neurologist based on Petersen's criteria [29]: (i) subjective complaint of decline in memory on self-or informant report, (ii) isolated memory deficits (determined by the 10-word list learning test $\leq$ 5) [30], (iii) essentially preserved general cognitive function (determined by the Mental State Examination T10 (MSET10) > 22 or adjusted for educational level) [31], (iv) largely intact activities of daily living using the clinical review with the patients and informant interview, and (v) clinical dementia (determined by the global Clinical Dementia Rating (CDR) scale < 1) [32]. Additionally, all participants with MCI were required to score on the Montreal Cognitive Assessment (MoCA) $\leq$ 24 to affirm the presence of subtle cognitive impairment [33]. The cognitively intact control group with similar age (plus or minus two years), gender, body mass index (plus or minus two units), and education level (plus or minus two years) was recruited from Chiang Mai community via local community groups and social media advertisements (e.g., Facebook). To be eligible, all participants in the control group were required to have intact cognitive function, as identified by the normal scores of the MSET10 and MoCA.

The inclusion criteria for both groups were being able to walk independently and safely without an assistive device for at least 10 meters and being able to comprehend and follow the instructions. The exclusion criteria were: (i) had major health conditions that affect walking ability or safely during testing such as musculoskeletal, neurological, cardiorespiratory problems or any unstable medical conditions (e.g., Parkinson's disease, stroke, peripheral vertigo, acute joint pain, uncorrected visual and/or hearing impairment), (ii) had severe dizziness when turning the head in horizontal and/or vertical directions, (iii) had depressive symptoms (determined by the Thai Geriatric Depression Scale-15 score > 6) [34], (iv) current use of sedative drugs, antipsychotic drugs, and narcotic drugs, and (v) had leg length discrepancy > 2.5 cm (assessed from the anterior superior iliac spine (ASIS) to the tip of medial malleolus in a standing position using a flexible tape measure) [35].

In this study, we assessed participants' decision-making capacity by evaluating their comprehension of the research study, including its purpose, procedures, risks, and benefits. Additionally, we appraised their capability to make a rational decision regarding participation in the study. All participants provided written informed consent prior to enrolling in the study.

The institution's ethical review committee for research in humans approved this consent procedure (approval number: AMSEC-65EX-025). The start and end of the study were from the 26th of August 2022 to the 9th of April 2023. Before gait testing, participants in both groups were interviewed about personal demographic data, including age, weight, height, education level, fall incidence within the past year, medical conditions, and medication used. Additionally, participants underwent an assessment of attention and executive functions, which were previously determined to be closely related to gait control, using the Digit Span Forward and Backward Tests and Trail Making Test Parts A and B, respectively [36,37].

## Assessment of gait smoothness under complex motor walking tasks

A triaxial accelerometer device (DynaPort MoveTest, McRoberts, The Hague, The Netherlands) with a sampling frequency of 100 Hz was used to measure the IH, which is an indicator of gait smoothness along three orthogonal axes, including anteroposterior (AP), mediolateral (ML), and vertical (VT) directions [38,39]. Following the standard protocol for accelerometer placement, the accelerometer was mounted at the middle of participants' fifth lumbar spine level and fastened with the supplied elastic belt [40] (Fig 1). In addition, The McRoberts logo on the device was placed outside; with this, the axes x, y, and z were automatically aligned with the direction of gravity. This placement allowed a dependable fixation of the accelerometer and supported the computation of trunk signal detection algorithm [41,42]. The trunk acceleration signal during walking was transmitted in real-time via Bluetooth to a software database on the personal computer, where the signal-based parameters including IH were subsequently analyzed by an algorithm developed by Micó-Amigo et al. [43].

After a triaxial accelerometer setting, participants were requested to walk along a 10-meter walkway at their comfortable speed for two trials to use as baseline data. As for the gait testing protocol, participants were evaluated for gait smoothness under four complex motor walking tasks on the outdoor walking terrain as follows:

a. Walking a narrow path (WN) (Fig 2A): Participants were asked to walk within the lines marked with tape on the floor. The wide path was set individually at 50% of the distance between each participant's ASIS [44].

b. Walking around an obstacle (WO) (Fig 2B): Participants were instructed to alternately walk around the cones on both the right and left sides (zig-zag patterns). Each cone was placed at a distance of 61 cm in width and 75 cm in length [45].

c. Walking straight with horizontal head turns (W-HH) (Fig 2C): Participants were asked to continue walking straight while turning their heads once to the right and then once to the left, respectively [46]. The assessor instructed the participants to "look right" or "look left" in consecutive order, every 1 meter.

d. Walking straight with vertical head turns (W-VH) (Fig 2D): Participants were requested to keep walking straight while alternately tipping their heads up and down. The assessor instructed the participants to "look up" or "look down" in consecutive order, every 1 meter.

For all walking tasks, participants were asked to walk at a self-selected speed over a 10-meter distance for 2 trials in each condition (for a total of 8 trials). All complex motor walking tasks were randomized. A 1-meter distance from the beginning and end of the walkway was excluded from the data analysis to discourage gait acceleration and deceleration events. Participants were given a demonstration before testing, and standard instructions were provided throughout the trials. A 1-minute resting period was allowed in each trial.

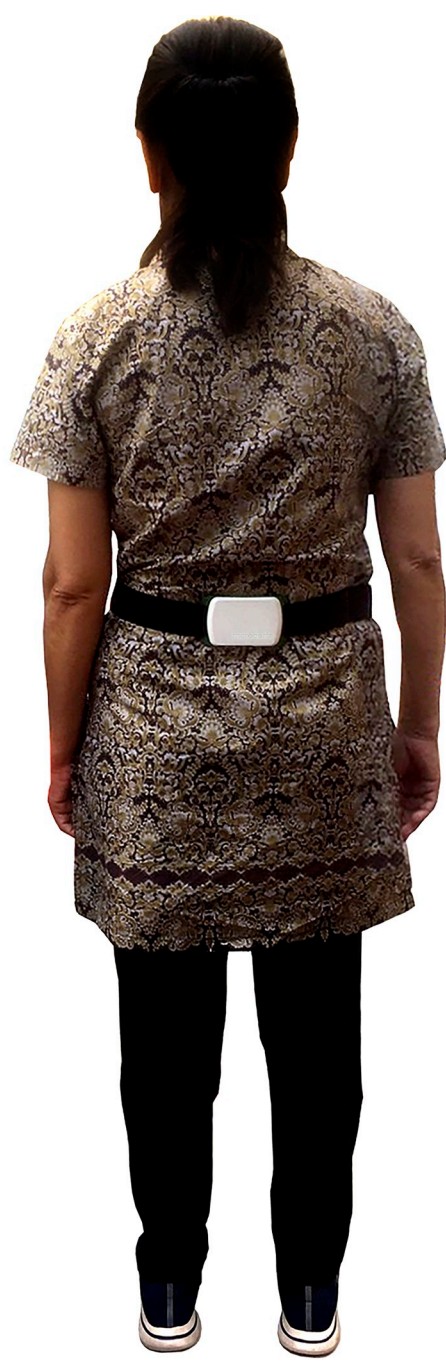

**Fig 1. Placement of triaxial accelerometer on the lower back at the level of the fifth lumbar vertebra.**

### Gait smoothness definition

Gait smoothness, expressed as the IH, is a metric of gait quality that determines the rhythmic intersegmental coordination of trunk and pelvis movement within a stride by exploiting the periodicity of the signal [20]. The IH was calculated from the power of the first six harmonics estimated through a discrete Fourier transform (DFT). As calculated by a DFT, $P_0$ is the power spectral density of the fundamental frequency (first harmonic) and $\sum_{i=0}^{5} P$ is the cumulative

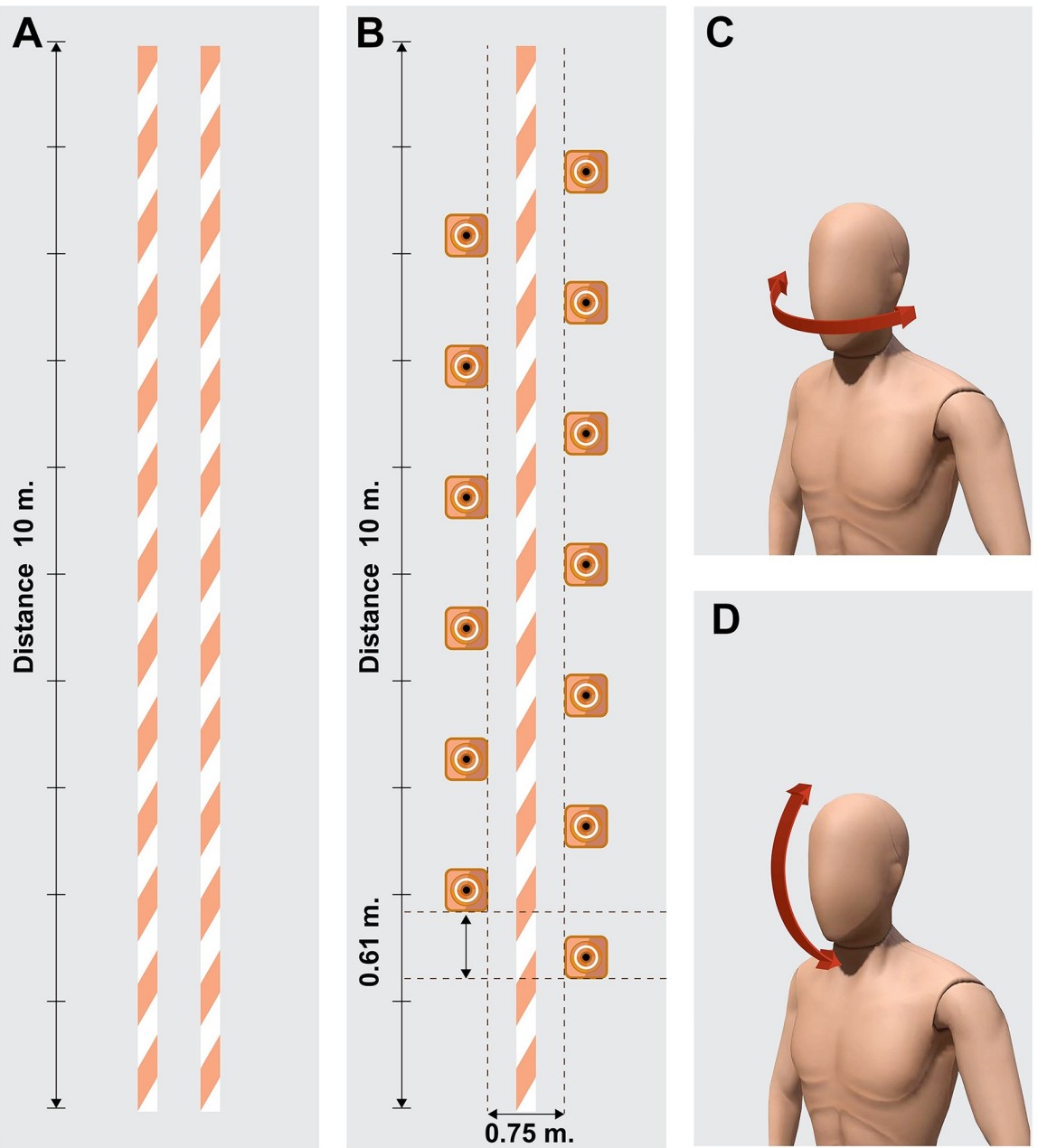

**Fig 2. Gait assessment under complex motor walking tasks.** (A) Walking a narrow path, (B) Walking around an obstacle, (C) Walking straight while the horizontal head turns, and (D) Walking straight while the vertical head turns.

sum of the powers of the fundamental frequency and the first five superharmonics (multiplication of the fundamental frequency) (Eqs 1–3). The value of IH showed numeric and unitless, in which a power ratio of 1 denotes perfect gait smoothness [20]. The IH measures while performing complex motor walking tasks were quantified into three directions of acceleration: AP, ML, and VT.

$$IH_{AP} = \frac{P_{0\_AP}}{\sum_{i=0}^{5} P\_AP_i}$$

(1)

$$IH_{ML} = \frac{P_{0\_ML}}{\sum_{i=0}^{5} P\_ML_i} \tag{2}$$

$$IH_{VT} = \frac{P_{0\_VT}}{\sum_{i=0}^{5} P\_VT_i} \tag{3}$$

## Statistical analysis

SPSS software for macOS version 26 (IBM, Armonk, NY) was used to perform the statistical analyses. An independent sample t-test (continuous data) and Chi-Square test (categorial data) were used to compare the participant demographic profiles between the MCI and control groups. The Shapiro-Wilk test was used to determine data normality. A multivariate analysis was employed to address the differences in IH (i.e., AP, ML, and VT) between the two groups for each complex motor walking task (i.e., WN, WO, W-HH, W-VH). A significance level was set at $p < 0.05$.

## Results

### Participant characteristics

The demographics, cognitive score tests, and baseline gait smoothness of the MCI and control groups are shown in Table 1. There were no significant differences between groups in demographic characteristics ($p > 0.05$). As for cognitive performance, the MCI group had lower scores on MoCA ($p = 0.001$) and Digit Span Test than the control group ($p = 0.01$). Gait speed and IH in all directions (i.e., AP, ML, and VT) at baseline assessment were similar between the two groups ($p > 0.05$).

### Gait smoothness during walking under complex motor tasks

The mean and standard deviation of the IH between groups for the WN, WO, W-HH and W-VH conditions are presented in Table 2 (S1 File). The multivariate analysis revealed a significant difference of the IH-ML in the W-HH condition ($p = 0.04$) between the two groups, in which the MCI group had a lower IH-ML value when compared to the control group. There were no significant differences in all other IH directions and other conditions between the two groups. Moreover, gait speed for each complex motor walking task was similar between the MCI and control groups.

## Discussion

Complex walking tasks that require a high demand of gait control may be used to identify early gait impairments in individuals with MCI. The IH, as a measure of gait smoothness, serves as a robust metric for the rhythmic intersegmental coordination of trunk and pelvis movements within a stride, contributing to gait stability. Currently, there are limited studies that investigate gait smoothness in people with MCI during challenging walking tasks relevant to real-life walking situations. The purpose of this preliminary study was to explore the potential for motor stress tests in detecting subtle gait impairment in older adults with MCI. To our knowledge, this study is possibly the first to investigate gait smoothness in older adults with MCI during complex motor tasks necessitating significant walking stability.

**Table 1. Participant characteristics in the control and MCI groups (N = 36).**

| Characteristics | Control (n = 18) | MCI (n = 18) | p-value[#] |
|---|---|---|---|
| Age (yr) | 67.72 ± 4.63 | 67.89 ± 4.64 | 0.92 |
| Gender (Male: Female, n) | 6:12 | 6: 12 | 1.00 |
| Body mass index (kg/m$^2$) | 24.84 ± 3.12 | 24.17 ± 2.92 | 0.51 |
| Education level (yr) | 15.89 ± 3.16 | 15.22 ± 3.44 | 0.55 |
| Number of medications | 1.17 ± 1.20 | 1.56 ± 1.15 | 0.33 |
| Number of falls in the past year | 0.22 ± 0.55 | 0.28 ± 0.75 | 0.80 |
| MSET10 (score, total score = 29) | 27.00 ± 1.24 | 26.67 ± 1.33 | 0.44 |
| MoCA (score, total score = 30) | 26.22 ± 1.52 | 22.00 ± 1.94 | 0.001[a] |
| DST (score, total score = 28) | 17.11 ± 3.45 | 14.39 ± 2.83 | 0.01[a] |
| TMT A (sec) | 45.74 ± 15.48 | 49.11 ± 18.76 | 0.56 |
| TMT B (sec) | 104.64 ± 38.14 | 121.14 ± 41.72 | 0.22 |
| TMT B-A (sec) | 58.90 ± 34.36 | 72.03 ± 40.72 | 0.30 |
| Baseline IH-AP | 0.68 ± 0.08 | 0.68 ± 0.10 | 0.96 |
| Baseline IH-ML | 0.75 ± 0.08 | 0.75 ± 0.10 | 0.98 |
| Baseline IH-VT | 0.85 ± 0.06 | 0.84 ± 0.05 | 0.46 |
| Baseline gait speed (m/s) | 1.11 ± 0.15 | 1.02 ± 0.20 | 0.12 |

All values are shown as mean ± standard deviation except for gender, number of medications, and falls in the past year.

[#] Independent sample t-test for continuous data and Chi-square test for categorical data.

[a] Significant difference between two groups, $p < 0.05$.

MSET10 = Mental State Examination T10; MoCA = Montreal Cognitive Assessment; DST = Digit Span Test; TMT A = Trail Making Test Part A; TMT B = Trail Making Test Part B; TMT B-A = subtracting Part B from Part A; IH-AP = Index of harmonicity in the anteroposterior direction; IH-ML = Index of harmonicity in the mediolateral direction; IH-VT = Index of harmonicity in the vertical direction.

Findings from this study demonstrated that the MCI group showed a decrease in gait smoothness in the mediolateral direction while walking with horizontal head turns compared to the peer group. During walking, the gravity sensors of the vestibular system, which are calibrated by the visual input, are responsible for gaze stabilization when the head moves via the vestibulo-ocular reflex [47]. In addition, head direction cells, which are a type of hippocampal neuron, utilize vestibular input to encode heading direction during movement through the vestibular-hippocampal pathway [48]. Previous studies found degeneration of the hippocampus and entorhinal cortex in the early stage of Alzheimer's disease, including MCI [49,50]. It is potentially possible that older adults with MCI experience a decline in the hippocampal formation, which is critical for spatial orientation and navigation; thereby potentially influencing the accuracy of current spatial location and directional heading in space. Therefore, the reduced smoothness of rhythmic trunk and pelvis movement in the mediolateral direction observed in older adults with MCI might indicate diminished postural balance control when encountering walking conditions that affect visual and vestibular information, such as horizontal head turns during walking. Singh et al. [51] reported that the horizontal head-turn walking task adversely affected gait performance in older persons with poor lateral balance compared to the head-forward walking task, suggesting that this walking condition requires a higher demand on mediolateral balance control. Our findings align with a previous study, suggesting that the horizontal head-turning load on the postural control system may lead to gait instability in older adults with MCI, as reflected in a reduction in mediolateral regulation. In the vertical head-turn condition, the results indicated a decrease in gait smoothness in the MCI group compared to the control group, reaching marginal significance (p = 0.06). To date, no studies have contrasted the difference in control of the gait between horizontal and vertical head-turn walking in older

**Table 2. The IH values of the MCI and control groups under complex motor walking tasks.**

| IH | Control group (n = 18) | MCI group (n = 18) | p-value[#] |
|---|---|---|---|
| **Walking a narrow path (WN)** | | | |
| IH-AP | 0.76 ± 0.07 | 0.74 ± 0.11 | 0.65 |
| IH-ML | 0.76 ± 0.09 | 0.73 ± 0.11 | 0.34 |
| IH-VT | 0.21 ± 0.14 | 0.23 ± 0.14 | 0.65 |
| Gait speed (m/s) | 0.97 ± 0.21 | 0.87 ± 0.23 | 0.17 |
| **Walking around an obstacle (WO)** | | | |
| IH-AP | 0.18 ± 0.12 | 0.20 ± 0.15 | 0.69 |
| IH-ML | 0.53 ± 0.14 | 0.49 ± 0.18 | 0.55 |
| IH-VT | 0.37 ± 0.19 | 0.36 ± 0.19 | 0.93 |
| Gait speed (m/s) | 0.27 ± 0.05 | 0.26 ± 0.05 | 0.82 |
| **Walking straight with horizontal head turns (W-HH)** | | | |
| IH-AP | 0.37 ± 0.23 | 0.36 ± 0.18 | 0.94 |
| IH-ML | 0.74 ± 0.12 | 0.64 ± 0.16 | 0.04[a] |
| IH-VT | 0.69 ± 0.16 | 0.61 ± 0.17 | 0.16 |
| Gait speed (m/s) | 0.88 ± 0.14 | 0.85 ± 0.16 | 0.55 |
| **Walking straight with vertical head turns (W-VH)** | | | |
| IH-AP | 0.61 ± 0.19 | 0.60 ± 0.18 | 0.90 |
| IH-ML | 0.70 ± 0.16 | 0.66 ± 0.15 | 0.44 |
| IH-VT | 0.73 ± 0.15 | 0.62 ± 0.15 | 0.06 |
| Gait speed (m/s) | 0.89 ± 0.13 | 0.84 ± 0.16 | 0.39 |

All values are shown as mean ± standard deviation.

[#] Multivariate analysis.

[a] Significant difference between two groups, p < 0.05.

IH = Index of harmonicity; IH-AP = Index of harmonicity in the anteroposterior direction; IH-ML = Index of harmonicity in the mediolateral direction; IH-VT = Index of harmonicity in the vertical direction.

adults with and without MCI. A previous study exhibited a significant correlation between gaze stability during vertical head movement and gait performance in patients with vestibular disorders (r = 0.88) [52]. This finding suggested that gaze stability performance, which is mediated via the vestibulo-ocular reflex, is prominent for successful walking during vertical head movement. Further research is warranted to examine this issue in older adults with MCI.

The present study also revealed that the smoothness of gait did not differ between the MCI and control groups when navigating around an obstacle and walking a narrow path, contrary to our hypothesis. Regarding walking around an obstacle, gait speed of the MCI and control groups was obviously slower than other walking conditions. A prior study found that gait speed influenced the smoothness of gait, with slow gait speeds affecting smoothness more than comfortable and fast speeds [19], as also evidenced in our findings. We postulated that a decreased gait speed while navigating obstacles might be due to older adults adopting a more cautious gait pattern to reduce the magnitudes of the trunk accelerations to ensure walking stability; however, although older people attempt to attenuate gait speed, this does not maintain gait stability (reflecting via a lower IH in all directions), which is in line with previous studies [28,53]. Therefore, continuous walking around the obstacles would be one of the motor stress tests that compromise gait stability in older adults with and without MCI. Furthermore, the current findings demonstrated that the gait smoothness during the narrow path walking condition was similar for both groups. It is conceivable that walking a narrow path did not pose a sufficient challenge to unveil subtle gait changes in individuals with MCI. Studies examining

gait smoothness concerning IH during narrow path walking among individuals with cognitive limitations compared to healthy older and young adults are limited, making it challenging to compare the findings.

There are certain limitations in this study that need to be addressed. Participants in the present study were healthy and well-functioning older adults. Thus, our findings may not be generalized to older adults with underlying physical conditions relevant to mobility problems. Moreover, we restricted the use of gait smoothness as an outcome measure due to the fact that it indicates a quality of rhythmic gait regulation; thereby, further studies that include other gait variables such as spatiotemporal, variability, and kinematics measures may provide insight into the whole picture of gait control in older adults with MCI during complex motor walking tasks. In addition to the triaxial lumbar accelerometer employed in the present study, adding another sensor at the thoracic level would provide a further understanding of how upper and lower intersegmental coordination movements contribute to gait stability. Moreover, a long walking distance that covers more strides would allow for a steady gait outcome analysis. Finally, the findings in this study are preliminary and should be interpreted with caution; thus, future studies are warranted to confirm these findings.

## Conclusions

The present exploratory study provides preliminary evidence that older adults with MCI have compromised mediolateral gait stability during horizontal head-turning while walking. Measurement of gait smoothness under this walking condition may allow early detection of gait changes in older adults with MCI.

## Supporting information

**S1 File. Gait smoothness while walking under complex motor walking tasks in the MCI and control groups.** Mean, standard deviation, maximum, and minimum values of index of harmonicity of AP, ML, and VT directions for each individual under each complex motor walking task: Walking a narrow path, walking around an obstacle, horizontal head turns while walking, and vertical head turns while walking.
(PDF)

## Acknowledgments

The authors thank Poosri N., Kumfu S., and Lonlue N. for their assistance throughout all data collection. We would also like to thank physical therapists and staff in the subdistrict administrative organization and the elderly school for their contributions to participant recruitment. Finally, we would like to thank all the participants for their time and willingness to participate.

## Author Contributions

**Conceptualization:** Thanpidcha Poosri, Sirinun Boripuntakul, Somporn Sungkarat, Teerawat Kamnardsiri, Atiwat Soontornpun, Kanokporn Pinyopornpanish.

**Data curation:** Thanpidcha Poosri, Sirinun Boripuntakul, Somporn Sungkarat.

**Formal analysis:** Thanpidcha Poosri, Sirinun Boripuntakul, Somporn Sungkarat.

**Funding acquisition:** Thanpidcha Poosri.

**Investigation:** Thanpidcha Poosri, Sirinun Boripuntakul, Atiwat Soontornpun, Kanokporn Pinyopornpanish.

**Methodology:** Thanpidcha Poosri, Sirinun Boripuntakul, Somporn Sungkarat, Teerawat Kamnardsiri, Atiwat Soontornpun, Kanokporn Pinyopornpanish.

**Project administration:** Thanpidcha Poosri, Sirinun Boripuntakul.

**Resources:** Thanpidcha Poosri.

**Software:** Teerawat Kamnardsiri.

**Supervision:** Sirinun Boripuntakul, Somporn Sungkarat.

**Validation:** Sirinun Boripuntakul.

**Visualization:** Teerawat Kamnardsiri.

**Writing – original draft:** Thanpidcha Poosri, Sirinun Boripuntakul.

**Writing – review & editing:** Thanpidcha Poosri, Sirinun Boripuntakul, Somporn Sungkarat, Teerawat Kamnardsiri, Atiwat Soontornpun, Kanokporn Pinyopornpanish.

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
