## [Decision Letter · Decision Letter 0]

20 Oct 2023

PONE-D-23-25030Gait smoothness during high-demand motor walking tasks in older adults with mild cognitive impairmentPLOS ONE

Dear Dr. Boripuntakul,

Thank you for submitting your manuscript to PLOS ONE. After careful consideration, we feel that it has merit but does not fully meet PLOS ONE’s publication criteria as it currently stands. Therefore, we invite you to submit a revised version of the manuscript that addresses the points raised during the review process.

We look forward to receiving your revised manuscript.

Kind regards,

Renato S. Melo, PhD

Academic Editor

PLOS ONE

2. Please describe in your methods section how capacity to provide consent was determined for the participants in this study. Please also state whether your ethics committee or IRB approved this consent procedure. If you did not assess capacity to consent please briefly outline why this was not necessary in this case.

Reviewers' comments:

Reviewer's Responses to Questions

**Comments to the Author**

1. Is the manuscript technically sound, and do the data support the conclusions?

Reviewer #1: Yes

Reviewer #2: No

Reviewer #3: Yes

2. Has the statistical analysis been performed appropriately and rigorously? 

Reviewer #1: I Don't Know

Reviewer #2: No

Reviewer #3: Yes

3. Have the authors made all data underlying the findings in their manuscript fully available?

Reviewer #1: Yes

Reviewer #2: Yes

Reviewer #3: Yes

4. Is the manuscript presented in an intelligible fashion and written in standard English?

Reviewer #1: Yes

Reviewer #2: Yes

Reviewer #3: Yes

5. Review Comments to the Author

Reviewer #1: This quasi-experimental study was well-conceived, and the manuscript is well-organized and written. I made a few comments on word choice and have several questions/need for additional information (see comments made directly on the .pdf).

Reviewer #2: Summary: The authors investigated gait smoothness in adults with and without mild cognitive impairment (MCI) during complex tasks that are common in daily life. The authors quantified gait smoothness during walking as the Index of Harmonicity (IH) of a lumbar-mounted accelerometer. The IH of adults with and without MCI were compared across a narrow path task, an obstacle avoidance task, and a head turning task. The authors found moderate differences only in the mediolateral IH during the heading turning task, concluding that the IH during a head turning task may provide a valuable outcome to be used as early fall risk detection in people with MCI.

The results in this study have not been published elsewhere, to my knowledge. The manuscript is intelligible and with only minor typos, though focusing the writing on a specific target impact would improve the manuscript’s clarity.

The authors provide strong mechanistic and empirical support for their premise: that experimental paradigms challenging motor and vestibular function would reveal subtle gait deficits in adults with (vs. without) MCI. However, the limited support for using the IH as a measure of fall risk and the limitations of the experimental methodology do not inspire confidence in the robustness of the primary outcomes or the knowledge gained from the study. Combined with limitations in the statistical analysis, these limitations provide poor support for the manuscript’s conclusions. I do not think that the study’s conclusions can be supported without significant additional data collection and revised analyses. Therefore, I do not recommend the manuscript for publication.

I have provided specific comments below.

1. The focus of the manuscript should be made more precise. The focus and implications seem to alternate between early fall risk detection, improving clinical outcomes, and characterizing stability deficits in adults with MCI.

2. I am concerned by conclusions drawn from a single significant result amid many comparisons. Because multiple comparisons would support the same hypothesis—that gait smoothness differentially changes in adults with versus without MCI—the authors should correct for multiple comparisons. Given the single significant result (p = 0.04), it is likely that no comparisons would be significant after correction.

3. The authors should justify the use of a lumbar-worn accelerometer and use of linear accelerations to compute the Index of Harmonicity (IH). I am concerned that a lumbar sensor is a poor estimator of trunk accelerations. It is certainly less sensitive than (e.g.,) a sensor mounted at the thoracic spine. A lumbar sensor will not detect bending mid-torso.

Further, I question the author’s interpretation of a lumbar sensor as measuring trunk accelerations. A lumbar sensor is likely a better estimate of pelvis acceleration, but a poor measure of both pelvis and lumbar accelerations. This could readily be amended by revising the manuscript’s language.

As an aside, the oft-cited paper credited with first using the index of harmonicity (IH) used the pelvis and thoracic angles [1], though there does not appear to be strong support for one sensor placement versus another. Subsequent papers using a single lumbar sensor do not seem to evaluate the sensitivity of sensor configuration on the index of harmonicity.

[1] Lamoth CJ, Beek PJ, Meijer OG. Pelvis–thorax coordination in the transverse plane during gait. Gait & posture. 2002 Oct 1;16(2):101-14.

4. The language should be more specific the manuscript. For example, gait stability is invoked as rationale for evaluating IH, but not strongly connected to IH of the pelvis/lumbar spine and is not discussed in the context of the results. Additional justification for why pelvis/lumbar smoothness represents gait stability would also improve the manuscript.

Further, the rationale for using the IH in the context of fall prevention is poor: the IH is not a reliable predictor of falls [1]. How, then, can the authors claim that the IH during a head turning task can be used for early fall detection without relating these results to fall risk?

[1] Riva F, Toebes MJ, Pijnappels MA, Stagni R, Van Dieën JH. Estimating fall risk with inertial sensors using gait stability measures that do not require step detection. Gait & posture. 2013 Jun 1;38(2):170-4.

5. The short walking trials used in this study are likely insufficient to extract stable measures of the IH [1]. Over one hundred strides are needed for the IH to approach a stable value in non-disabled adults. How can the authors trust their outcomes with only 20m worth of strides of data per subject-trial? At a minimum, a bootstrapping procedure is needed to evaluate how cycle-to-cycle variations in gait and random sampling impact their results.

[1] Riva F, Bisi MC, Stagni R. Gait variability and stability measures: Minimum number of strides and within-session reliability. Computers in biology and medicine. 2014 Jul 1;50:9-13.

6. The authors should specify if the trial order was randomized.

7. The authors should consider how instructions for the head turning task may impact IH. Per the instructions, the rate of head turning would differ based on each participant’s walking speed (“head turn every 1 meter” [sic]). As gaze stabilization performance is associated with walking performance in adults with vestibular dysfunction [1] and head turning alters gait stability [2], it is plausible that varying the rate of head rotation would differentially alter gait stability and/or smoothness in people with different walking speeds.

[1] Whitney SL, Marchetti GF, Pritcher M, Furman JM. Gaze stabilization and gait performance in vestibular dysfunction. Gait Posture. 2009 Feb;29(2):194-8. doi: 10.1016/j.gaitpost.2008.08.002. Epub 2008 Sep 23. PMID: 18815040; PMCID: PMC4879829.

[2] Fitzgerald, C., Thomson, D., Zebib, A. et al. A comparison of gait stability between younger and older adults while head turning. Exp Brain Res 238, 1871–1883 (2020). https://doi.org/10.1007/s00221-020-05846-3

8. I am surprised to see such small differences in Trails B-A scores between groups. In these tasks, I might expect set-shifting ability to correlate with changes in IH. The authors may see larger effects of head turning or obstacle negotiation if groups differed in clinical scores associated with set-shifting or motor-cognitive integration.

9. In general, the Discussion is not strongly focused on interpreting and contextualizing the study’s findings. For example, paragraph 2 could be written almost entirely without conducting the study. Condensing the Discussion and improving focus on contextualizing results would improve the manuscript.

Reviewer #3: The paper provides valuable insights into the assessment of gait changes in older adults with Mild Cognitive Impairment (MCI) under complex motor tasks. Here is a review of the introduction and methodology sections:

Introduction Review:

1. The introduction sets the stage effectively by discussing the importance of executive function, attention, and their connection to safe mobility.

2. The rationale for the study, focusing on the potential for gait abnormalities in individuals with MCI, is well-explained. This establishes the significance of the research.

3. The introduction highlights the existing research and points out discrepancies, which helps justify the need for the study.

4. The introduction clearly defines terms like MCI, making it reader-friendly for those not well-versed in the subject matter.

5. The study's objectives and hypotheses are clearly stated, providing a solid foundation for the research.

Methodology Review:

1. The methodology section adequately describes the participant selection process and sample size calculation, contributing to the paper's transparency.

2. Inclusion and exclusion criteria are well-defined, ensuring that the study focuses on the intended population.

3. Ethical considerations and approval from the institution's review committee are appropriately mentioned, underlining the ethical aspects of the research.

4. The assessment of gait smoothness under complex motor tasks is well-explained, including the use of a triaxial accelerometer device, making it clear to the reader how data was collected.

5. The statistical analysis section is detailed and provides a clear understanding of how the data was analyzed.

6. The results are presented in a clear and organized manner, with tables that effectively summarize the findings.

General Review:

1. The paper is generally well-structured and follows a logical flow, with each section building upon the previous one.

2. The paper maintains a strong focus on the research question throughout the introduction and methodology sections.

3. The language is technical but clear, making the content accessible to a broader audience.

4. The limitations are acknowledged, and the need for further research is suggested, showing the author's awareness of the study's boundaries.

In summary, the introduction effectively introduces the research problem and its significance, while the methodology provides a clear framework for the study, including participant selection and data collection methods. The paper appears to be well-structured and technically sound, making it a valuable contribution to the field of research on gait abnormalities in older adults with MCI.

6. PLOS authors have the option to publish the peer review history of their article (what does this mean?). If published, this will include your full peer review and any attached files.

Reviewer #1: No

Reviewer #2: No

Reviewer #3: No

---

## [Author Response · Author response to Decision Letter 0]

9 Dec 2023

Response to Reviewers’ Comments

Journal Requirements:

Comment # 1. Please ensure that your manuscript meets PLOS ONE's style requirements, including those for file naming. The PLOS ONE style templates can be found at https://journals.plos.org/plosone/s/file?id=wjVg/PLOSOne_formatting_sample_main_body.pdf and https://journals.plos.org/plosone/s/file?id=ba62/PLOSOne_formatting_sample_title_authors_affiliations.pdf

Response # 1. We have formatted our manuscript according to PLOS ONE's style requirements.

Comment # 2. Please describe in your methods section how capacity to provide consent was determined for the participants in this study. Please also state whether your ethics committee or IRB approved this consent procedure. If you did not assess capacity to consent please briefly outline why this was not necessary in this case. 

Response # 2. We have included this information in the methods section as follows (page 7, line 150-155): 

“In this study, we assessed participants' decision-making capacity by evaluating their comprehension of the research study, including its purpose, procedures, risks, and benefits. Additionally, we appraised their capability to make a rational decision regarding participation in the study. All participants provided written informed consent prior to enrolling in the study. The institution’s ethical review committee for research in humans approved this consent procedure (approval number: AMSEC-65EX-025).”

Reviewers' comments:

Reviewer's Responses to Questions

Reviewer # 1: 

This quasi-experimental study was well-conceived, and the manuscript is well-organized and written. I made a few comments on word choice and have several questions/need for additional information (see comments made directly on the .pdf).

Response: Thank you very much. 

We have revised the words and sentences throughout the manuscript according to the reviewer’s suggestion as follows:

Page 3, line 50: We have revised “resulting” to “result”.

Page 6, line 118: We have revised “walking while the vertical head turns” to “walking while making vertical head turns”.

Page 7, line 142: We have revised “safety” to “safely”.

Page 9, line 192 and 196: We have revised “walking in concurrent with horizontal/vertical head turns” to “walking straight with horizontal/vertical head turns”.

Page 15, line 295: We have revised “in mediolateral direction” to “in the mediolateral direction”.

Page 16, line 311: We have revised “high” to “higher”.

Page 16, line 315: We have revised “a marginal” to “marginal”.

Page 17, line 335: We have revised “the narrow walking” to “the narrow path walking”.

Comment # 1. Please provide references attesting to the accuracy (validity) and precision (reliability) of this device.

Response # 1. We have included the references about the validity and reliability of the triaxial accelerometer device (DynaPort MoveMonitor, McRoberts) in the methods section (page 8, line 168).

Comment # 2. Line 166: please spell this abbreviation out and put IH in parentheses. Response # 2. We spelled out the full name and abbreviation of IH on page 4, line 88. After that, we use the “IH” throughout the manuscript to keep the format consistent.

Comment # 3. Please describe any calibration procedure that might be used for this device prior to data collection.

Response # 3. The triaxial accelerometer device (DynaPort MoveMonitor, McRoberts) used in the present study has been calibrated prior to being dispatched (1). We followed the standardized procedures from previous studies to ensure the accuracy of gait measurement (1-2). 

We have included the information in the methods section as follows (page 8, line 168-174):

"Following the standard protocol for accelerometer placement, the accelerometer was mounted at the middle of participants’ fifth lumbar spine level and fastened with the supplied elastic belt [40] (Fig 1). In addition, The McRoberts logo on the device was placed outside; with this, the axes x, y, and z were automatically aligned with the direction of gravity. This placement allowed a dependable fixation of the accelerometer and supported the computation of trunk signal detection algorithm [41-42]” 

References:

1. McRoberts [Internet]. Netherlands: McRoberts BV; 2023 [updated 2023; cited 2023 Dec 1]. Available from: https://www. https://www.mcroberts.nl/

2. van Hees V, Slootmaker S, Groot G, Mechelen W, Van Lummel R. Reproducibility of a triaxial seismic accelerometer (DynaPort). Med Sci Sports Exerc. 2009;41:810-7. doi: 10.1249/MSS.0b013e31818ff636

3. Mazzà C, Alcock L, Aminian K, Becker C, Bertuletti S, Bonci T, et al. Technical validation of real-world monitoring of gait: a multicentric observational study. BMJ Open. 2021;11(12):e050785. doi: 10.1136/bmjopen-2021-050785.

Comment # 4. Line 217: should be "IH is..."

Response # 4. We have rechecked and it should be “Po is….”. We aimed to define the formula for IH calculation. 

Comment # 5. Line 219: I assume that this measure is unitless. If so, please declare this or provide the units of the measure.

Response # 5. Yes, it is unitless. We have mentioned this in the text (page 10, line 219-220).

Comment # 6. Line 222: suggest: "antero-posterior (AP), medio-lateral (ML), and vertical (VT)

Response # 6. We spelled out the full name and abbreviation of anteroposterior, mediolateral, and vertical on page 8, line 167-168. After that, we use the “AP”, “ML”, and “VT” throughout the manuscript to keep the format consistent. 

Comment # 7. Please define P and Po in the part of gait smoothness definition.

Response # 7. We have defined P and Po (page 10, line 217-219).

Comment # 8. In the statistical analysis, should a correction be made with multiple t-tests, e.g., Bonferonni?

Response # 8. As this is an exploratory analysis, no adjustment was made for multiple comparisons of outcomes. However, as the dependent variables (i.e., IH in the AP, ML, and VT directions) of each walking task are related, the statistician suggested that a multivariate analysis should be conducted instead of an independent t-test. The results are still the same, wherein the MCI group had a significantly lower IH in the ML direction compared to the control group under horizontal head turns while walking (p = 0.04).

We have revised the statistical analysis, and acknowledged in the study limitation that findings are preliminary.

Comment # 9. In the discussion part, prior to restating the study purpose please briefly state the rationale/need for the study.

Response # 9. We have added a brief explanation of the rationale in the discussion part as follows (page 15, line 285-290): 

“Complex walking tasks that require a high demand of gait control may be used to identify early gait impairments in individuals with MCI. The IH, as a measure of gait smoothness, serves as a robust metric for the rhythmic intersegmental coordination of trunk and pelvis movements within a stride, contributing to gait stability. Currently, there are limited studies that investigate gait smoothness in people with MCI during challenging walking tasks relevant to real-life walking situations.”

Comment # 10. Line 317-319: be careful - correlation is not cause/effect. You should report the precise correlation value and let the reader decide magnitude because I have seen r values of 0.3 which were statistically significant and might be called strong, when surely this number may not be clinically relevant, i.e., a r = 0.3 is an r2 = 0.09 which is relatively meaningless.

Response # 10. Thank you for pointing this out. We have revised this sentence as the reviewer’s suggestion (page 16, line 319-321). 

Reviewer # 2: 

Summary: The authors investigated gait smoothness in adults with and without mild cognitive impairment (MCI) during complex tasks that are common in daily life. The authors quantified gait smoothness during walking as the Index of Harmonicity (IH) of a lumbar-mounted accelerometer. The IH of adults with and without MCI were compared across a narrow path task, an obstacle avoidance task, and a head turning task. The authors found moderate differences only in the mediolateral IH during the heading turning task, concluding that the IH during a head turning task may provide a valuable outcome to be used as early fall risk detection in people with MCI.

The results in this study have not been published elsewhere, to my knowledge. The manuscript is intelligible and with only minor typos, though focusing the writing on a specific target impact would improve the manuscript’s clarity.

The authors provide strong mechanistic and empirical support for their premise: that experimental paradigms challenging motor and vestibular function would reveal subtle gait deficits in adults with (vs. without) MCI. However, the limited support for using the IH as a measure of fall risk and the limitations of the experimental methodology do not inspire confidence in the robustness of the primary outcomes or the knowledge gained from the study. Combined with limitations in the statistical analysis, these limitations provide poor support for the manuscript’s conclusions. I do not think that the study’s conclusions can be supported without significant additional data collection and revised analyses. Therefore, I do not recommend the manuscript for publication. I have provided specific comments below.

Comment # 1. The focus of the manuscript should be made more precise. The focus and implications seem to alternate between early fall risk detection, improving clinical outcomes, and characterizing stability deficits in adults with MCI.

Response # 1. We thank the reviewer for the thorough assessment and valuable comments on the manuscript. We apologize for the confusion caused by the writing that appears to alternate the study’s focus back and forth. The focus of this study was to identify gait stability deficits in older adults with MCI. Specifically, we aimed to explore subtle MCI-related changes in gait under a high-demand motor walking task by using gait smoothness as a measurement indicator. Findings in this study might offer early detection of gait deficits and facilitate timely gait intervention for older adults with MCI. We have revised the introduction, objective, and conclusion based on this comment and the comment #2 (page 3, line 51-52; page 5, line 98-102; page 18, line 355-358).

Comment # 2. I am concerned by conclusions drawn from a single significant result amid many comparisons. Because multiple comparisons would support the same hypothesis-that gait smoothness differentially changes in adults with versus without MCI- the authors should correct for multiple comparisons. Given the single significant result (p = 0.04), it is likely that no comparisons would be significant after correction.

Response # 2. We appreciate the important points that the reviewer raises. This study aimed to explore which complex walking task(s) could differentiate gait changes between individuals with MCI and cognitively intact controls. As this is an exploratory analysis, no adjustment was made for multiple comparisons of outcomes. Each complex walking task (i.e., walking a narrow path, walking around an obstacle, horizontal head turns while walking, and vertical head turns while walking) was established to be a high-demand motor task that requires a unique and independent underlying mechanism of gait control. Further, as the dependent variables (i.e., IH in the AP, ML, and VT directions) of each walking task are related, the statistician suggested that a multivariate analysis should be conducted instead of an independent t-test. The results are still the same, wherein the MCI group had a significantly lower IH in the ML direction compared to the control group under horizontal head turns while walking (p = 0.04).

We have revised the introduction, statistical analysis, acknowledged in the study limitation that findings are preliminary and should be interpreted with caution, and moderated the conclusion accordingly.

Comment # 3. The authors should justify the use of a lumbar-worn accelerometer and use of linear accelerations to compute the Index of Harmonicity (IH). I am concerned that a lumbar sensor is a poor estimator of trunk accelerations. It is certainly less sensitive than (e.g.,) a sensor mounted at the thoracic spine. A lumbar sensor will not detect bending mid-torso.

Further, I question the author’s interpretation of a lumbar sensor as measuring trunk accelerations. A lumbar sensor is likely a better estimate of pelvis acceleration, but a poor measure of both pelvis and lumbar accelerations. This could readily be amended by revising the manuscript’s language.

As an aside, the oft-cited paper credited with first using the index of harmonicity (IH) used the pelvis and thoracic angles [1], though there does not appear to be strong support for one sensor placement versus another. Subsequent papers using a single lumbar sensor do not seem to evaluate the sensitivity of sensor configuration on the index of harmonicity.

[1] Lamoth CJ, Beek PJ, Meijer OG. Pelvis-thorax coordination in the transverse plane during gait. Gait & posture. 2002 Oct 1;16(2):101-14.

Response # 3. Previous studies have indicated that accelerometers can be placed either in the lower or upper trunk to measure trunk acceleration signals (1-4). In this study, we chose to mount an accelerometer at the lumbar because a systematic review has shown that trunk acceleration measured by a single triaxial accelerometer over the lower back level is a valid and reliable method for gait metrics analysis (4). Moreover, the single lumbar accelerometer used in the present study (DynaPort Hybrid, McRoberts, Netherlands) has also been valid and reliable for gait assessment across various populations (5-9). One explanation is that the lower back level approximates the body’s center of mass, which is the internal representation of the whole-body movement during walking (10). Thus, the selection of a single reference point on the lower trunk is based on the assumption that the task of maintaining stability when walking primarily requires controlling the motion of the center of mass. However, we agree with the reviewer that adding another sensor, such as at the upper trunk level, may provide more insight into the role and pattern of trunk movement control contributing to gait stability. We have included the above information in the study limitation as follows (Page 18, line 347-349):

“In addition to the triaxial lumbar accelerometer employed in the present study, adding another sensor at the thoracic level would provide a further understanding of how upper and lower intersegmental coordination movements contribute to gait stability.”

As the reviewer mentioned, a previous study by Lamoth et al. (11) did not strongly support one sensor placement over another. In their study, thoracic and pelvis axial rotations during walking were measured using standard method as light-emitting diodes (LEDs) and the Selspot™ Camera System (Selcom). Afterward, Lamoth et al. established the spectral analysis method for assessing the coordination of thoracic and pelvis movement within a stride using the IH, in which a power ratio of 1 indicates that the intersegmental movement is perfectly harmonic. These findings indicate that the IH can reflect rhythmic axial intersegmental coordination during walking, potentially contributing to overall gait stability.

Regarding the evidence of the sensitivity of sensor configuration on the IH involves using a single lumbar sensor. Unfortunately, there is currently no study assessing the sensitivity of the single body-fixed sensor for IH, as this gait metric is gaining attention and is still an ongoing area of research. Nevertheless, a previous study revealed that the single lumbar accelerometer was a valid, reliable, and sensitive instrument for measuring spatiotemporal gait parameters in older adults (12). In addition, a systematic review has reported that single-point lower back accelerometers were acceptable for assessing dynamic gait outcomes, including the IH (4). 

References:

1. Doi T, Hirata S, Ono R, Tsutsumimoto K, Misu S, Ando H. The harmonic ratio of trunk acceleration predicts falling among older people: results of a 1-year prospective study. J Neuroeng Rehabil. 2013;28:10:7. 

2. Terrier P, Reynard F. Effect of age on the variability and stability of gait: a cross-sectional treadmill study in healthy individuals between 20 and 69 years of age. Gait Posture. 2015;41:170-4.

3. Menz HB, Lord SR, Fitzpatrick RC. Acceleration patterns of the head and pelvis when walking are associated with risk of falling in community-dwelling older people. J Gerontol A Biol Sci Med Sci. 2003;58(5):M446-52. 

4. Mobbs RJ, Perring J, Raj SM, Maharaj M, Yoong NKM, Sy LW, et al. Gait metrics analysis utilizing single-point inertial measurement units: a systematic review. Mhealth. 2022;8:9. 

5. Houdijk H, Appelman FM, Van Velzen JM, van der Woude LH, van Bennekom CA. Validity of DynaPort GaitMonitor for assessment of spatiotemporal parameters in amputee gait. J Rehabil Res Dev. 2008;45:1335-42.

6. Bautmans I, Jansen B, Van Keymolen B, Mets T. Reliability and clinical correlates of 3D-accelerometry based gait analysis outcomes according to age and fall-risk. Gait Posture. 2011;33:366-72.

7. de Bruin ED, Hubli M, Hofer P, Wolf P, Murer K, Zijlstra W. Validity and reliability of accelerometer-based gait assessment in patients with diabetes on challenging surfaces. J Aging Res. 2012;2012:954378.

8. Hartmann A, Luzi S, Murer K, de Bie RA, de Bruin ED. Concurrent validity of a trunk tri-axial accelerometer system for gait analysis in older adults. Gait Posture. 2009;29:444-8.

9. Hartmann A, Murer K, de Bie RA, de Bruin ED. Reproducibility of spatio-temporal gait parameters under different conditions in older adults using a trunk tri-axial accelerometer system. Gait Posture. 2009;30:351-5.

10. Yang F, Pai YC. Can sacral marker approximate center of mass during gait and slip-fall recovery among community-dwelling older adults? J Biomech. 2014;47(16):3807-12. 

11. Lamoth CJ, Beek PJ, Meijer OG. Pelvis-thorax coordination in the transverse plane during gait. Gait Posture. 2002;16(2):101-14.

12. Werner C, Heldmann P, Hummel S, Bauknecht L, Bauer JM, Hauer K. Concurrent validity, test-retest reliability, and sensitivity to change of a single body-fixed sensor for gait analysis during rollator-assisted walking in acute geriatric patients. Sensors (Basel). 2020;20(17):4866.

Comment # 4. The language should be more specific the manuscript. For example, gait stability is invoked as rationale for evaluating IH, but not strongly connected to IH of the pelvis/lumbar spine and is not discussed in the context of the results. Additional justification for why pelvis/lumbar smoothness represents gait stability would also improve the manuscript.

Further, the rationale for using the IH in the context of fall prevention is poor: the IH is not a reliable predictor of falls [1]. How, then, can the authors claim that the IH during a head turning task can be used for early fall detection without relating these results to fall risk?

[1] Riva F, Toebes MJ, Pijnappels MA, Stagni R, Van Dieën JH. Estimating fall risk with inertial sensors using gait stability measures that do not require step detection. Gait & posture. 2013 Jun 1;38(2):170-4. 

Response # 4. Thank you for the suggestion. We have incorporated the connection between lumbar/pelvis movement, the IH, and gait stability into the rationale, as suggested by the reviewer. Furthermore, we have discussed these aspects in the context of the results within the discussion section (page 4-5, line 87-94; page 16, line 305-307).

We agree with the reviewer that the use of IH in the context of fall detection is poor. Therefore, we removed sentences about IH and early fall risk detection throughout the manuscript. 

Comment # 5. The short walking trials used in this study are likely insufficient to extract stable measures of the IH [1]. Over one hundred strides are needed for the IH to approach a stable value in non-disabled adults. How can the authors trust their outcomes with only 20 m worth of strides of data per subject-trial? At a minimum, a bootstrapping procedure is needed to evaluate how cycle-to-cycle variations in gait and random sampling impact their results.

[1] Riva F, Bisi MC, Stagni R. Gait variability and stability measures: Minimum number of strides and within-session reliability. Computers in biology and medicine. 2014 Jul 1;50:9-13.

Response # 5. Several works have found that trunk acceleration-based measures are reliable for quantifying spatiotemporal gait parameters among healthy older adults and individuals with pathologic conditions (1-2); however, apart from Riva et al. (3), very limited studies investigate the reliability of trunk accelerometers on gait smoothness measures, including the IH. Moreover, no study has assessed cycle-to-cycle variations in the IH among older adults, both with and without cognitive limitations, especially under complex walking tasks.

As the reviewer’s concern regarding the stable measures of the IH, we further explored the consistency of IH between the first and second trials for each walking condition in both the MCI and control groups, using test-retest reliability. The results showed good to excellent test-retest reliability of the IH for all walking conditions (as shown in Table 1). 

Table 1 The intra-class correlation coefficient of the IH

Walking condition Control group MCI group

 IH (Mean ± SD) ICC 3,1 (95% CI) p-value IH (Mean ± SD) ICC 3,1 (95% CI) p-value

 1st trial 2nd trial 1st trial 2nd trial 

Walking forward (baseline) 

IH-AP 0.70 ± 0.08 0.67 ± 0.08 0.921 0.0001* 0.70 ± 0.12 0.67 ± 0.11 0.902 0.0001*

IH-ML 0.75 ± 0.09 0.75 ± 0.08 0.800 0.001* 0.75 ± 0.12 0.73 ± 0.12 0.908 0.0001*

IH-VT 0.86 ± 0.05 0.84 ± 0.07 0.913 0.0001* 0.85 ± 0.05 0.83 ± 0.05 0.898 0.0001*

Walking a narrow path

IH-AP 0.76 ± 0.07 0.75 ± 0.08 0.855 0.0001* 0.74 ± 0.12 0.74 ± 0.11 0.928 0.0001*

IH-ML 0.77 ± 0.09 0.75 ± 0.10 0.820 0.0001* 0.74 ± 0.10 0.71 ± 0.12 0.879 0.0001*

IH-VT 0.22 ± 0.14 0.20 ± 0.14 0.930 0.0001* 0.24 ± 0.14 0.23 ± 0.14 0.960 0.0001*

Walking around an obstacle

IH-AP 0.17 ± 0.10 0.20 ± 0.15 0.858 0.0001* 0.20 ± 0.15 0.20 ± 0.15 0.956 0.0001*

IH-ML 0.53 ± 0.15 0.52 ± 0.15 0.835 0.0001* 0.52 ± 0.17 0.49 ± 0.19 0.959 0.0001*

IH-VT 0.39 ± 0.20 0.34 ± 0.21 0.837 0.0001* 0.37 ± 0.19 0.35 ± 0.19 0.963 0.0001*

Horizontal head turns while walking

IH-AP 0.38 ± 0.23 0.35 ± 0.24 0.961 0.0001* 0.38 ± 0.21 0.34 ± 0.19 0.761 0.003*

IH-ML 0.73 ± 0.12 0.74 ± 0.12 0.958 0.0001* 0.63 ± 0.16 0.64 ± 0.17 0.974 0.0001*

IH-VT 0.67 ± 0.18 0.70 ± 0.16 0.899 0.0001* 0.59 ± 0.19 0.62 ± 0.16 0.898 0.0001*

Vertical head turns while walking

IH-AP 0.60 ± 0.19 0.62 ± 0.20 0.968 0.0001* 0.60 ± 0.19 0.60 ± 0.18 0.926 0.0001*

IH-ML 0.69 ± 0.16 0.71 ± 0.16 0.947 0.0001* 0.66 ± 0.15 0.66 ± 0.15 0.953 0.0001*

IH-VT 0.72 ± 0.18 0.73 ± 0.14 0.919 0.0001* 0.63 ± 0.16 0.62 ± 0.16 0.915 0.0001*

*p-value < 0.05

ICC = intra-class correlation coefficient

IH = Index of harmonicity; IH-AP = Index of harmonicity in the anteroposterior direction; IH-ML = Index of harmonicity in the mediolateral direction; IH-VT = Index of harmonicity in the vertical direction

The results from Table 1 indicate an acceptable reliability of the IH measures using trunk accelerometers obtained from short walking trials. Currently, several studies determine gait smoothness, including the IH and Harmonic Ratio (HR), in older adults and individuals with pathologic conditions within a short distance range from 8 to 30 meters (4-7). However, we agree with the reviewer that adding the number of strides by increasing the walking distance would allow for stable gait outcomes. We acknowledged that it is a limitation of the study and stated in the manuscript (page 18, line 349-350) as follows:

“Moreover, a long walking distance that covers more strides would allow for a steady gait outcome analysis.”

References:

1. Kobsar D, Charlton JM, Tse CTF, Esculier JF, Graffos A, Krowchuk NM, et al. Validity and reliability of wearable inertial sensors in healthy adult walking: a systematic review and meta-analysis. J. Neuroeng Rehabil. 2020;17:62.

2. Werner C, Heldmann P, Hummel S, Bauknecht L, Bauer JM, Hauer K. Concurrent validity, test-retest reliability, and sensitivity to change of a single body-fixed sensor for gait analysis during rollator-assisted walking in acute geriatric patients. Sensors (Basel). 2020;20(17):4866. 

3. Riva F, Bisi MC, Stagni R. Gait variability and stability measures: minimum number of strides and within-session reliability. Comput Biol Med. 2014;50:9-13.

4. Simonetti E, Pillet H, Vannozzi G, Loiret I, Villa C, Bascou J, et al. Investigating symmetry in amputee gait through the improved harmonic ratio: influence of the stride segmentation method. Comput Methods Biomech Biomed Engin. 2019;22(sup1):S221-S3.

5. Micó-Amigo ME, Kingma I, Heinzel S, Nussbaum S, Heger T, van Lummel RC, et al. Dual vs. single tasking during circular walking: what better reflects progression in Parkinson's disease? Front Neurol. 2019;10:372.

6. Pau M, Mulas I, Putzu V, Asoni G, Viale D, Mameli I, et al. Smoothness of gait in healthy and cognitively impaired individuals: a study on Italian elderly using wearable inertial sensor. Sensors (Basel). 2020;20(12):3577.

7. Javadpour S, Sinaei E, Salehi R, Zahednejad S, Motealleh A. Comparing the effects of single-task versus dual-task balance training on gait smoothness and functional balance in community-dwelling older adults: a randomized controlled trial. J Aging Phys Act. 2022;30(2):308-15.

Comment # 6. The authors should specify if the trial order was randomized.

Response # 6. Thank you. The trial order was randomized. We have added this statement in the method section (page 10, line 206-207).

Comment # 7. The authors should consider how instructions for the head turning task may impact IH. Per the instructions, the rate of head turning would differ based on each participant’s walking speed (“head turn every 1 meter” [sic]). As gaze stabilization performance is associated with walking performance in adults with vestibular dysfunction [1] and head turning alters gait stability [2], it is plausible that varying the rate of head rotation would differentially alter gait stability and/or smoothness in people with different walking speeds.

[1] Whitney SL, Marchetti GF, Pritcher M, Furman JM. Gaze stabilization and gait performance in vestibular dysfunction. Gait Posture. 2009 Feb;29(2):194-8. doi: 10.1016/j.gaitpost.2008.08.002. Epub 2008 Sep 23. PMID: 18815040; PMCID: PMC4879829.

[2] Fitzgerald C, Thomson D, Zebib A. et al. A comparison of gait stability between younger and older adults while head turning. Exp Brain Res. 238, 1871-1883 (2020). https://doi.org/10.1007/s00221-020-05846-3

Response # 7. Thank you for your point of view. We agree with the reviewer that the rate of head turning while walking relates to walking speed, which may result in gait stability. In the present study, we consistently gave a standard instruction for participants to turn their heads as fast and as far as possible in response to the command every 1 meter. With this, there was no discernible difference in the head turning rate between individuals or walking trials. As we found, all participants could follow the instructions and complete the walking tasks without obviously changing gait speed from baseline. Moreover, the results showed that gait speed in head turning conditions was similar between the MCI and control groups, suggesting that the difference in the IH between the two groups was not influenced by gait speed (Table 2 in the manuscript).

Comment # 8. I am surprised to see such small differences in Trails B-A scores between groups. In these tasks, I might expect set-shifting ability to correlate with changes in IH. The authors may see larger effects of head turning or obstacle negotiation if groups differed in clinical scores associated with set-shifting or motor-cognitive integration.

Response # 8. We agree with the reviewer that larger effects of head turning or obstacle negotiation might be observed if Trail B-A scores were differences between the two groups. The head turning or obstacle negotiation while walking serves as a motor dual-tasking as participants have to allocate their attention between walking and a head turning or obstacle negotiation task. Non-significant differences in TMT B-A scores in the present study are likely due to a large variation of the data. 

Comment # 9. In general, the Discussion is not strongly focused on interpreting and contextualizing the study’s findings. For example, paragraph 2 could be written almost entirely without conducting the study. Condensing the Discussion and improving focus on contextualizing results would improve the manuscript.

Response # 9. We appreciate your constructive feedback. We have re-written the discussion to focus on interpretating and contextualizing the study findings. Thank you once again for your valuable suggestion. 

Reviewer # 3: 

The paper provides valuable insights into the assessment of gait changes in older adults with Mild Cognitive Impairment (MCI) under complex motor tasks. Here is a review of the introduction and methodology sections:

Comment # 1. 

Introduction Review:

The introduction sets the stage effectively by discussing the importance of executive function, attention, and their connection to safe mobility. The rationale for the study, focusing on the potential for gait abnormalities in individuals with MCI, is well-explained. This establishes the significance of the research. The introduction highlights the existing research and points out discrepancies, which helps justify the need for the study. The introduction clearly defines terms like MCI, making it reader-friendly for those not well-versed in the subject matter. The study's objectives and hypotheses are clearly stated, providing a solid foundation for the research.

Response # 1. Thank you very much.

Comment # 2. 

Methodology Review:

The methodology section adequately describes the participant selection process and sample size calculation, contributing to the paper's transparency. Inclusion and exclusion criteria are well-defined, ensuring that the study focuses on the intended population. Ethical considerations and approval from the institution's review committee are appropriately mentioned, underlining the ethical aspects of the research. The assessment of gait smoothness under complex motor tasks is well-explained, including the use of a triaxial accelerometer device, making it clear to the reader how data was collected. The statistical analysis section is detailed and provides a clear understanding of how the data was analyzed. The results are presented in a clear and organized manner, with tables that effectively summarize the findings.

Response # 2. Thank you very much.

Comment # 3. General Review:

The paper is generally well-structured and follows a logical flow, with each section building upon the previous one. The paper maintains a strong focus on the research question throughout the introduction and methodology sections. The language is technical but clear, making the content accessible to a broader audience. The limitations are acknowledged, and the need for further research is suggested, showing the author's awareness of the study's boundaries.

In summary, the introduction effectively introduces the research problem and its significance, while the methodology provides a clear framework for the study, including participant selection and data collection methods. The paper appears to be well-structured and technically sound, making it a valuable contribution to the field of research on gait abnormalities in older adults with MCI.

Response # 3. Thank you very much.

---

## [Decision Letter · Decision Letter 1]

12 Dec 2023

PONE-D-23-25030R1Gait smoothness during high-demand motor walking tasks in older adults with mild cognitive impairmentPLOS ONE

Dear Dr. Boripuntakul,

Thank you for submitting your manuscript to PLOS ONE. After careful consideration, we feel that it has merit but does not fully meet PLOS ONE’s publication criteria as it currently stands. Therefore, we invite you to submit a revised version of the manuscript that addresses the points raised during the review process.

We look forward to receiving your revised manuscript.

Kind regards,

Renato S. Melo, PhD

Academic Editor

PLOS ONE

Journal Requirements:

Reviewers' comments:

Reviewer's Responses to Questions

**Comments to the Author**

1. If the authors have adequately addressed your comments raised in a previous round of review and you feel that this manuscript is now acceptable for publication, you may indicate that here to bypass the “Comments to the Author” section, enter your conflict of interest statement in the “Confidential to Editor” section, and submit your "Accept" recommendation.

Reviewer #1: All comments have been addressed

2. Is the manuscript technically sound, and do the data support the conclusions?

Reviewer #1: Yes

3. Has the statistical analysis been performed appropriately and rigorously? 

Reviewer #1: Yes

4. Have the authors made all data underlying the findings in their manuscript fully available?

Reviewer #1: Yes

5. Is the manuscript presented in an intelligible fashion and written in standard English?

Reviewer #1: Yes

6. Review Comments to the Author

Reviewer #1: Thank you for addressing my comments. I have two additional comments that are relatively minor (see pdf)

7. PLOS authors have the option to publish the peer review history of their article (what does this mean?). If published, this will include your full peer review and any attached files.

Reviewer #1: No

---

## [Author Response · Author response to Decision Letter 1]

14 Dec 2023

Reviewer # 1: 

Comment # 1. Should consider stating "may lead..." since walking smoothness does not necessarily conflate with stability, does it? Furthermore, you did not report that participants lost balance during their walks suggesting that they were not unstable.

Response # 1: Thank you for the suggestion. We have revised the words according to the reviewer’s suggestion (Page 16, line 312).

Comment # 2. Your findings suggest that older healthy groups should be compared to young healthy groups with regard to IH.

Response # 2: Thank you for pointing this out. We have revised the sentence as the reviewer’s suggestion as follows (Page 17, line 337-340):

“Studies examining gait smoothness concerning IH during narrow path walking among individuals with cognitive limitations compared to healthy older and young adults are limited, making it challenging to compare the findings.”

---

## [Decision Letter · Decision Letter 2]

18 Dec 2023

Gait smoothness during high-demand motor walking tasks in older adults with mild cognitive impairment

PONE-D-23-25030R2

Dear Dr. Boripuntakul,

We’re pleased to inform you that your manuscript has been judged scientifically suitable for publication and will be formally accepted for publication once it meets all outstanding technical requirements.

Kind regards,

Renato S. Melo, PhD

Academic Editor

PLOS ONE

Additional Editor Comments (optional):

Reviewers' comments:

Reviewer's Responses to Questions

**Comments to the Author**

1. If the authors have adequately addressed your comments raised in a previous round of review and you feel that this manuscript is now acceptable for publication, you may indicate that here to bypass the “Comments to the Author” section, enter your conflict of interest statement in the “Confidential to Editor” section, and submit your "Accept" recommendation.

Reviewer #1: All comments have been addressed

2. Is the manuscript technically sound, and do the data support the conclusions?

Reviewer #1: Yes

3. Has the statistical analysis been performed appropriately and rigorously? 

Reviewer #1: Yes

4. Have the authors made all data underlying the findings in their manuscript fully available?

Reviewer #1: Yes

5. Is the manuscript presented in an intelligible fashion and written in standard English?

Reviewer #1: Yes

6. Review Comments to the Author

Reviewer #1: All concerns/comments/questions/suggestions have been addressed. There is a solid introduction with identification of a need for the study, explicit purpose statement and conclusion that is consistent with the primary purpose and results. The methods are sufficiently detailed so that this study could be replicated.

7. PLOS authors have the option to publish the peer review history of their article (what does this mean?). If published, this will include your full peer review and any attached files.

Reviewer #1: No

---

## [Editor Report · Acceptance letter]

10 Jan 2024

PONE-D-23-25030R2 

PLOS ONE

Dear Dr. Boripuntakul, 

I'm pleased to inform you that your manuscript has been deemed suitable for publication in PLOS ONE. Congratulations! Your manuscript is now being handed over to our production team.

Kind regards, 

on behalf of

Dr. Renato S. Melo 

Academic Editor

PLOS ONE